# Comparison of Simple RNA Extraction Methods for Molecular Diagnosis of Hepatitis C Virus in Plasma

**DOI:** 10.3390/diagnostics12071599

**Published:** 2022-06-30

**Authors:** Sayamon Hongjaisee, Yosita Jabjainai, Suthasinee Sakset, Kanya Preechasuth, Nicole Ngo-Giang-Huong, Woottichai Khamduang

**Affiliations:** 1Research Institute for Health Sciences, Chiang Mai University, Chiang Mai 50200, Thailand; sayamon.ho@cmu.ac.th; 2School of Medical Sciences, University of Phayao, Phayao 56000, Thailand; yosita.uk1703@gmail.com; 3Infectious Diseases Research Unit, Department of Medical Technology, Faculty of Associated Medical Sciences, Chiang Mai University, Chiangmai 50200, Thailand; jaa.suthasinee@gmail.com (S.S.); kanya.p@cmu.ac.th (K.P.); 4Maladies Infectieuses et Vecteurs: Écologie, Génétique, Évolution et Contrôle (MIVEGEC), Agropolis University Montpellier, Centre National de la Recherche Scientifique (CNRS), Institut de Recherche Pour le Développement (IRD), 34394 Montpellier, France; nicole.ngo-giang-huong@phpt.org; 5Associated Medical Sciences (AMS)-PHPT Research Collaboration, Chiang Mai 50200, Thailand

**Keywords:** RNA extraction, HCV, molecular diagnosis, RT-LAMP, RT-PCR, plasma

## Abstract

Nucleic acid extraction from biological samples is an important step for hepatitis C virus (HCV) diagnosis. However, such extractions are mostly based on silica-based column methodologies, which may limit their application for on-site diagnosis. A simple, rapid, and field-deployable method for RNA extraction is still needed. In this study, we evaluated the efficacy of four simple RNA extraction methods for the detection of HCV in plasma samples: a silica-membrane-based method, a magnetic-beads-based method, boiling with diethyl pyrocarbonate (DEPC)-treated distilled water, and using a commercial lysis buffer. HCV RNA was detected using both real-time reverse transcription polymerase chain reaction (RT-PCR) and reverse transcription loop-mediated isothermal amplification (RT-LAMP). Using real-time RT-PCR, extracted RNA from the silica-membrane-based and magnetic-beads-based methods had a 100% detection rate for RNA extraction from plasma. Using RT-LAMP, extracted RNA from the silica-membrane-based method showed a 66% detection rate, while the magnetic-beads-based method had a 62% detection rate. In summary, magnetic-beads-based extraction can be used as an alternative RNA extraction method for on-site HCV detection. Boiling with DEPC-treated distilled water was not appropriate for low HCV load samples, and boiling with a lysis buffer was not recommended.

## 1. Introduction

Hepatitis C virus (HCV) is a leading cause of chronic liver disease, cirrhosis, and liver cancer. The current oral direct-acting antiviral (DAA) combinations can cure 95–99% patients [1], but the majority of HCV-infected people are unaware of their infection status. Testing for antibodies to HCV (anti-HCV) is insufficient to diagnose a current HCV infection or ongoing HCV replication. HCV RNA testing is needed. Real-time reverse transcription polymerase chain reaction (RT-PCR) is commonly used for HCV RNA detection/quantification in clinical practice. However, this test is rarely available for an on-site diagnosis, especially in remote settings because it requires specific reagents and instruments with high costs. The development of reverse transcription loop-mediated isothermal amplification assay (RT-LAMP) can facilitate the access to molecular testing for HCV. Indeed, the clinical sensitivity and specificity of RT-LAMP has been previously reported to be as high as 90–100% for HCV detection [2,3,4]. RT-LAMP also stands out in terms of rapidity, simplicity, cost-effectiveness, and accessibility, making it ideal for field or point-of-care use in remote settings where sophisticated and expensive equipment is not available. However, this technique still requires an effective nucleic acid extraction and purification method. Commercial nucleic acid extraction kits are currently based on a silica-based or on magnetic-beads-based technologies. The choice of utility depends on the settings and the number of samples to be tested daily. However, these extraction systems are expensive and require specific instruments such as a centrifuge or automated extractor, limiting the use of RT-LAMP for on-site diagnosis or in the field. In this study, we selected four simple extraction methods to evaluate their efficiency in combination with RT-LAMP detection: (1) a silica-membrane-based extraction method which combines the selective binding properties of a silica-based membrane with the speed of microspin technology; (2) a magnetic-beads-based extraction method which uses the reversible adsorption of nucleic acids to paramagnetic beads and a magnetic separator to clean and purify the nucleic acids; (3) a boiling plasma method; and (4) boiling plasma with a lysis buffer. The 2 boiling methods allow the rapid release of viral RNA. The results confirmed the presence of HCV RNA in extracted samples with real-time RT-PCR.

## 2. Materials and Methods

This study used leftover plasma samples of 50 HCV-infected individuals who had undergone HCV viral load testing and genotyping for clinical care at the Faculty of Associated Medical Sciences, Chiang Mai University. HCV viral load was initially measured using a commercial real-time RT-PCR assay (COBAS AmpliPrep/COBAS TaqMan HCV Test). The study was approved by the Human Experimentation Committee (Number 17/64, 2 July 2021) and the Institutional Biosafety Committee (Number CMUIBC0363004, 23 December 2020) of the Research Institute for Health Sciences, Chiang Mai University, Thailand. Viral RNA was extracted from plasma samples using four different methods: (1) NucleoSpin RNA Virus (Macherey-Nagel, Düren, Germany) and (2) NucleoMag Virus kit (Macherey-Nagel, Düren, Germany), following the manufacturer’s recommendations; (3) boiling with water, in which 100 µL plasma was mixed 1:1 with diethyl pyrocarbonate (DEPC)-treated distilled water (Invitrogen, CA, USA) and boiled at 95 °C for 10 min [5], the mixture was centrifuged at 8000× *g* for 1 min, and the supernatant was collected; and (4) boiling with lysis buffer, which was conducted as above, but 100 µL plasma was mixed 1:1 with a commercial lysis buffer (Lysis Buffer RAV1, Macherey-Nagel, Germany). The yield and purity of RNA extracted by the four methods were measured using a NanoDrop™ 2000/2000c Spectrophotometer. For RNA detection by real-time RT-PCR, four µL of viral nucleic acid extract was amplified with 400 nM each of HCV forward and reverse primers, as well as 100 nM of HCV probe [6] of the sensiFAST Probe Lo-ROX One-Step kit. Amplification was performed on an Applied Biosystems 7500 instrument as follows: 45 °C for 10 min; 95 °C for 2 min; 40 cycles of 95 °C for 5 s and 60 °C for 20 s. The fluorescence signal was measured at 60 °C for each cycle. For RNA amplification by RT-LAMP, five µL of nucleic acid extract was used as previously described [2]. Briefly, RT-LAMP reaction was processed at 65 °C for 60 min. The result was visualized with the naked eye based on the color change of the reaction mixture induced by pre-added hydroxynaphthol blue. The samples that turned sky blue were considered to be positive, while those that remained purple were considered to be negative. To evaluate the analytical sensitivity of the four extraction methods, 10-fold serial dilutions of plasma with HCV viral load of 10^6^ IU/mL in 1× phosphate buffer saline were prepared. Aliquots of each dilution were extracted by the four extraction methods. All RNA extracts were tested for HCV RNA by both real time RT-PCR and RT-LAMP. Each reaction was performed in triplicate.

## 3. Results

The efficacy of the four extraction methods was evaluated using serial dilutions of plasma with HCV viral load of 10^6^ IU/mL. The dilutions that were used were 10^5^, 10^4^, and 10^3^ IU/mL. The four extraction methods showed considerably variable quantities of RNA. Boiling plasma samples with a commercial lysis buffer provided the highest yield of RNA, with a concentration of 695.37 ng/µL, and boiling with DEPC-treated distilled water produced the lowest yield of 0 ng/µL. The purity ranged from 0.14 to 3.46 and was the highest with the silica-membrane-based extraction method (Table 1). Analytical sensitivity on triplicate of serial dilutions of HCV RNA extracts from silica-based membrane and magnetic beads were all detected by real-time RT-PCR with a similar cycle threshold (*C*_t_) and by RT-LAMP at the 10^6^ concentration. The numbers of triplicates detected by both techniques decreased when initial concentrations were 10^5^ or lower. RNA extracts obtained after boiling in DEPC-treated distilled water were less well detected at higher *C*_t_ than the silica-membrane and magnetic-bead RNA extracts. Though boiling with a commercial lysis buffer gave the highest RNA yield, none of the RNA extracts yielded a signal in real-time RT-PCR, regardless of the initial HCV RNA concentration. The results from RT-LAMP could not be interpreted, as the color turned from purple to sky blue immediately after adding the extracted RNA to the reaction mix. Thus, boiling with a commercial lysis buffer was not further evaluated.

Real-time RT-PCR detection of extract of plasma samples using silica-membrane-based or magnetic-beads-based methods provided positive results in 50/50 samples (100%), Table 2 and Appendix A. The RNA extracted from boiling with DEPC-treated distilled water had a 76% detectable rate (38/50). Using RT-LAMP for HCV detection, RNA extracted with the silica-membrane-based method showed the best results with a 66% detectable rate (33/50), while the magnetic-beads-based method showed a 62% rate (31/50). Boiling samples with DEPC-treated distilled water provided the least number of samples with HCV RNA detected; only 7 of 50 (14%) tested positive (Table 2 and Appendix A). RT-LAMP results according to the RNA extraction methods are shown in Appendix A. The analysis of correlation between the *C*_t_ values of RNA extracted from the three different extraction methods showed an excellent correlation between the *C*_t_ values of samples extracted with the silica-membrane-based and magnetic-beads-based methods (R^2^ = 0.88, Figure 1A). The correlation was less good between the *C*_t_ values of samples extracted using the silica-membrane-based and boiling with DEPC-treated distilled water methods (R^2^ = 0.60, Figure 1B).

## 4. Discussion

Our study showed variable sensitivities of molecular HCV detection in plasma samples depending on the RNA extraction method used. The silica-membrane-based extraction method is a simple bind–wash–elute process. As noted, the eluates of this kit contain both viral nucleic acids and carrier RNA, which may exceed the amount of authentic nucleic acids from the virus when quantified by the photometric method. However, the extracted RNA from this method is suitable for HCV molecular diagnosis by real-time RT-PCR, and by RT-LAMP when using samples with a high viral load. Previous studies also showed that this method was suitable for RNA isolation from plasma and ready for use in subsequent reactions for viral detection [7,8]. The spin-column-based method also efficiently extracted DNA and facilitated the amplification of targets by LAMP and PCR [9]. However, this extraction method is fast but is the most expensive of the four extraction methods tested. Furthermore, it requires a centrifuge for the separation steps. 

The magnetic-beads-based extraction method is based on the reversible adsorption of nucleic acids to paramagnetic beads and uses only a magnetic separator. It does not need a centrifugation step but requires careful pipetting to remove the solution from the beads. In this method, 200 µL plasma sample was used, compared to only 100–150 µL of sample in the other three methods, which may lead to an increase in the RNA extraction capacity. Previous studies also recommended the magnetic bead technology for viral RNA extraction from serum or plasma [10,11]. Our results showed that Ct values from HCV samples extracted by the magnetic-beads-based method and silica-membrane-based method correlated well. Thus, it can be used as an alternative extraction method. A previous study also showed that magnetic-bead extracted DNA had high agreement with spin-column extracted DNA for downstream LAMP testing [12]. However, samples with low viral loads may cause a false negative result when used with RT-LAMP detection. Boiling is a simple and rapid method to release viral RNA from samples, taking about 15 min. Previous studies suggested that simple or direct boiling without any additional purification steps can be used as an alternative RNA isolation method to detect viral infections in clinical samples [5,13]. However, this method yielded the lowest RNA concentration and purity, as compared to others. This might be explained by a degradation of RNA during boiling and/or the lack of an additional step for concentration. Plasma samples showed a slightly decreased sensitivity when processed by boiling prior to amplification. When using real-time RT-PCR for detection, the *C*_t_ values from HCV samples extracted by boiling were slightly higher than those of RNA extracted by silica-membrane-based and magnetic beads-based extraction methods and did not strongly correlate. Although the boiling method could be used as a cost-effective alternative to expensive extraction methods, it can only be used when samples have a high viral load. Boiling samples with a commercial lysis buffer provided the highest RNA concentration but a quite low purity of RNA. This might be due to the RNA carrier contained in the lysis buffer and the absence of an additional step to remove the lysis buffer components or purification prior to amplification. We were unable to detect any HCV RNA in all RNA extracts from this method, and the results of RT-LAMP could not be interpreted. This may be due to an effect of inhibitors created or released by boiling. Thus, boiling plasma samples with a commercial lysis buffer cannot be used for RNA extraction in viral detection. However, some studies suggested that boiling techniques could become a suitable DNA extraction method when LAMP assays are used for detection, but this may require specific reagents, such as Triton X-100 or Chelex buffer [14,15]. The limitation of this study may be the relatively low number of samples and the diversity of HCV genotypes tested.

## 5. Conclusions

In summary, this study demonstrated that the magnetic-beads-based extraction method can be used as an alternative method of plasma RNA extraction for HCV detection. This method is simple, rapid, and inexpensive, and it does not require a centrifugation process, which makes it suitable for on-site diagnosis or in the field when combined with the RT-LAMP technique. This approach may improve the accessibility of HCV testing and help to identify HCV-infected individuals in need of treatment.

## Figures and Tables

**Figure 1 diagnostics-12-01599-f001:**
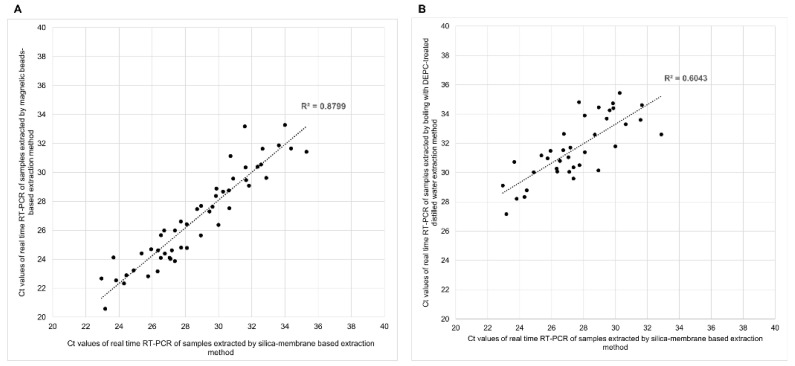
Correlation analysis between cycle threshold (*C*_t_) values obtained by real-time RT-PCR using extracted RNA (*n* = 50) from silica-membrane-based and magnetic-beads-based extraction methods (**A**) and boiling using DEPC-treated distilled water extraction methods (**B**).

**Table 1 diagnostics-12-01599-t001:** Efficacy of four different extraction methods for molecular diagnosis of HCV in plasma.

Plasma HCV RNA Level (IU/mL)		Extraction Methods			
		Silica-Membrane Based	Magnetic Beads-Based	Boiling with DEPC-Treated Distilled Water	Boiling with a Commercial Lysis Buffer
**10^6^**	Concentration (ng/µL)	345.80 ± 11.44	6.87 ± 4.03	125.63 ± 16.91	695.37 ± 13.62
	Purity (260/280)	3.45 ± 0.04	1.57 ± 0.12	0.87 ± 0.02	1.01 ± 0.04
	Real time RT-PCR, *C*t ^a^ (mean ± SD)	24.91 ± 0.29	24.11 ± 1.17	29.86 ± 0.77	Undetectable
	RT-LAMP, no. of detectable samples (%)	3/3 (100)	3/3 (100)	2/3 (67)	Uninterpreted
**10^5^**	Concentration (ng/µL)	347.73 ± 9.20	1.97 ± 0.83	28.30 ± 34.75	636.37 ± 1.19
	Purity (260/280)	3.44 ± 0.01	1.78 ± 0.90	0.99 ± 0.28	1.07 ± 0.04
	Real time RT-PCR, *C*t (mean ± SD)	28.27 ± 1.25	27.42 ± 0.78	32.48 ± 0.98	Undetectable
	RT-LAMP, no. of detectable samples (%)	1/3 (33)	1/3 (33)	0/3 (0)	Uninterpreted
**10^4^**	Concentration (ng/µL)	326.90 ± 47.56	2.80 ± 0.87	0.20 ± 0.14	620.47 ± 10.98
	Purity (260/280)	3.43 ± 0.04	1.97 ± 0.45	1.85 ± 2.06	1.08 ± 0.01
	Real time RT-PCR, *C*t (mean ± SD)	32.84 ± 0.42	32.80 ± 1.18	34.87 ± 1.01	Undetectable
	RT-LAMP, no. of detectable samples (%)	0/3 (0)	0/3 (0)	0/3 (0)	Uninterpreted
**10^3^**	Concentration (ng/µL)	318.33 ± 26.13	2.57 ± 0.72	0 ± 0.10	551.37 ± 126.27
	Purity (260/280)	3.46 ± 0.01	2.48 ± 0.73	0.14 ± 0.90	1.10 ± 0.01
	Real time RT-PCR, *C*t (mean ± SD)	Undetectable	Undetectable	Undetectable	Undetectable
	RT-LAMP, no. of detectable samples (%)	0/3 (0)	0/3 (0)	0/3 (0)	Uninterpreted

^a^*C*_t_: Cycle threshold.

**Table 2 diagnostics-12-01599-t002:** Clinical efficacy evaluation of the viral RNA extraction methods for HCV detection using real time RT-PCR and RT-LAMP.

HCV Viral Load (log_10_ IU/mL)	Extraction Methods					
	Silica-Membrane Based		Magnetic Beads-Based		Boiling with DEPC-Treated Distilled Water	
	Real Time RT-PCR, *C*_t_ (Mean ± SD)	RT-LAMP, No. of Detectable Samples (%)	Real Time RT-PCR, *C*_t_ (Mean ± SD)	RT-LAMP, No. of Detectable Samples (%)	Real Time RT-PCR, *C*_t_ (Mean ± SD)	RT-LAMP, No. of Detectable Samples (%)
6.01–7.00 (*N* = 10)	24.95 ± 1.64	10/10 (100)	23.19 ± 1.35	10/10 (100)	29.50 ± 1.27	3/10 (30)
5.01–6.00 (*N* = 14)	26.76 ± 1.13	13/14 (93)	24.59 ± 0.90	12/14 (86)	30.86 ± 0.93	4/14 (29)
4.01–5.00 (*N* = 14)	29.89 ± 1.37	9/14 (64)	27.89 ± 1.03	9/14 (64)	33.83 ± 1.07	0/14 (0)
3.01–4.00 (*N* = 12)	32.60 ± 1.44	1/12 (8)	31.21 ± 1.23	0/12 (0)	34.11 ± 0.72	0/12 (0)
Total (*N* = 50)	50 (100)	33 (66)	50 (100)	31 (62)	38 (76)	7 (14)

## Data Availability

Not applicable.

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
