# Peer review of "Comparison of Simple RNA Extraction Methods for Molecular Diagnosis of Hepatitis C Virus in Plasma"

_diagnostics, 2022, doi:10.3390/diagnostics12071599_

Round 1

Reviewer 1 Report

Comparison of simple RNA extraction methods for molecular diagnosis of hepatitis C virus in plasma.

Diagnostics. MDPI.

The article evaluates four nucleic acid extraction methods, especially RNA, as a first step for the molecular detection of hepatitis C virus. The authors use four extraction methods (silica-based columns, magnetic beads, boiling with DEPC-treated distilled water and boiling with a commercial lysis buffer). Subsequently, the amplification is carried out by conventional RT-PCR and by RT-LAMP.

Introduction. The authors state that the main extraction methods are based on silica-based columns (lines 50-52). This may be true in the realm of authors, but in many countries that’s not true, especially if we are talking about automated extraction, the priority method of extraction is by means of magnetic particles (Boom).

Results. In whole, boiling with lysis buffer yielded the highest RNA concentration, but apparently this method later causes technical problems in amplification, both by PCR and LAMP, and then was nor further evaluated. Membrane-based methods reach the highest purity, while boiling with DEPC-treated distilled water did not work at all.

The authors should make it clear in this first paragraph (lines 88 to 107), it refers exclusively to the controls with different viral loads, and not to the real samples. In any case, the study shows that the methods based on centrifugation with columns and on magnetic beads work better than the rest, and that, in any case, RT-PCR offers a significantly higher sensitivity than RT-LAMP when the loads viral are low. All these data indicate that the limitation of using LAMP as an alternative to PCR in settings with low technological resources is that it is a less sensitive method when viral loads are low, and that this limitation is exacerbated by the fact that that the most readily available methods in these areas extract worse, and therefore increase this difference in sensitivity. Overall, the study is well designed and carried out. The biggest problem that I find is that it is limited to corroborating data that was already known, such as the differences in efficiency in the extraction of the different methods or the greater sensitivity of RT-PCR compared to LAMP.

Reviewer 2 Report

The paper presented for review, "Comparison of Simple RNA Extraction Methods for Molecular Diagnosis of Hepatitis C Virus in Plasma", compares four methods of RNA isolation for molecular HCV diagnostics. I believe that the discussed topic is to some extent innovative and certainly provides interesting information that will be interesting and possible for other researchers to use. The work is written well, the results are correctly and aesthetically presented. Literature adequately cited. After the corrections, I encourage to publish in Diagnostics.

Major revision:

The discussion included the principles of operation and methodological descriptions of individual isolation methods. This is missing in the introduction, which would allow the reader to become familiar with the topic. It would be good to move these descriptions. The discussion should be supplemented with more detailed comparisons of other scientists' research and the formulation of appropriate conclusions.

Round 2

Reviewer 1 Report

Now I consider the paper might be tready for publication, should the editor consider the content is interesting enough for readers.

Reviewer 2 Report

Thank you very much for your answer. It is satisfactory.